# Is There an Optimal Combination of AREDS2 Antioxidants Zeaxanthin, Vitamin E and Vitamin C on Light-Induced Toxicity of Vitamin A Aldehyde to the Retina?

**DOI:** 10.3390/antiox11061132

**Published:** 2022-06-09

**Authors:** Małgorzata B. Różanowska, Barbara Czuba-Pełech, Bartosz Różanowski

**Affiliations:** 1School of Optometry and Vision Sciences, Cardiff University, Cardiff CF24 4HQ, Wales, UK; 2Cardiff Institute for Tissue Engineering and Repair (CITER), Cardiff University, Cardiff CF24 4HQ, Wales, UK; 3Department of Biophysics, Faculty of Biochemistry, Biophysics and Biotechnology, Jagiellonian University, 30-387 Kraków, Poland; barbara.czuba-pelech@uj.edu.pl; 4Institute of Biology, Pedagogical University, 30-084 Kraków, Poland; bartosz.rozanowski@up.krakow.pl

**Keywords:** carotenoid, xanthophyll, zeaxanthin, α-tocopherol, ascorbate, retina, retinal pigment epithelium, photosensitized oxidation, phototoxicity, age-related macular degeneration

## Abstract

Vitamins C and E and zeaxanthin are components of a supplement tested in a large clinical trial—Age-Related Eye Disease Study 2 (AREDS2)—and it has been demonstrated that they can inhibit the progression of age-related macular degeneration. The aim of this study was to determine the optimal combinations of these antioxidants to prevent the phototoxicity mediated by vitamin A aldehyde (ATR), which can accumulate in photoreceptor outer segments (POS) upon exposure to light. We used cultured retinal pigment epithelial cells ARPE-19 and liposomes containing unsaturated lipids and ATR as a model of POS. Cells and/or liposomes were enriched with lipophilic antioxidants, whereas ascorbate was added just before the exposure to light. Supplementing the cells and/or liposomes with single lipophilic antioxidants had only a minor effect on phototoxicity, but the protection substantially increased in the presence of both ways of supplementation. Combinations of zeaxanthin with α-tocopherol in liposomes and cells provided substantial protection, enhancing cell viability from ~26% in the absence of antioxidants to ~63% in the presence of 4 µM zeaxanthin and 80 µM α-tocopherol, and this protective effect was further increased to ~69% in the presence of 0.5 mM ascorbate. The protective effect of ascorbate disappeared at a concentration of 1 mM, whereas 2 mM of ascorbate exacerbated the phototoxicity. Zeaxanthin or α-tocopherol partly ameliorated the cytotoxic effects. Altogether, our results suggest that the optimal combination includes upper levels of zeaxanthin and α-tocopherol achievable by diet and/or supplementations, whereas ascorbate needs to be at a four-fold smaller concentration than that in the vitreous. The physiological relevance of the results is discussed.

## 1. Introduction

Vitamin A aldehyde, also known as all-*trans*-retinal or all-*trans*-retinaldehyde (ATR), can transiently accumulate in the retina as a result of the absorption of light by visual pigments [1,2,3,4,5]. Light absorption by visual pigments causes the photoisomerization of its chromophore 11-*cis*-retinaldehyde, which is bound to the amine group of Lysine 296 via protonated Schiff-base linkage. This leads to the activation of the visual pigment to its biochemically active state, where it can bind the G protein transducin, thereby initiating the phototransduction cascade, leading to visual perception. Subsequently, the isomerized chromophore, ATR, is hydrolyzed from the protein. ATR has potent photosensitizing properties: upon photoexcitation with UV or blue light, it forms an excited triplet state which can transfer the excess energy to molecular oxygen to generate an excited state of oxygen—singlet oxygen [1,2,3,6,7,8,9,10]. It has been determined that, depending on its environment, up to 30% of absorbed photons can be used for the generation of singlet oxygen by ATR [11,12]. It has been also reported that ATR can generate free radicals such as superoxide [12]. Both singlet oxygen and superoxide are reactive oxygen species that can exert damage to surrounding biomolecules of the photoreceptor outer segment and neighboring RPE cells [1,2,3,6,7,8,9,10].

While there are enzymes responsible for the clearance of ATR, mainly ATP-binding transporter rim protein (ABCR, also known as ABCA4) and retinol dehydrogenase 8 (RDH8), their velocity as well as the availability of co-factors, such as ATP and NADPH, are limited, and mutations in genes coding these enzymes can further limit their activities [1,2,3,4,6,13,14,15]. Therefore, a transient accumulation of ATR may occur after exposure to bright light of the dark-adapted retina [1,2,3,4,16,17,18,19,20]. Certain genetic mutations causing dysfunction of ABCR are responsible for increasing the susceptibility of such retina to light-induced injury shown in animals and are associated with development of Stargardt’s disease and a subset of cases of other blinding diseases, retinitis pigmentosa and age-related macular degeneration (AMD) in humans [4,15,16]. All these retinal diseases—AMD, Stargardt’s disease and retinitis pigmentosa—are associated with increased oxidative stress and oxidative damage in the outer retina, comprised of photoreceptive neurons and retinal pigment epithelium (RPE) [21,22,23,24,25,26,27,28,29,30].

Photoreceptors and RPE contain a number of antioxidants, including three antioxidants of dietary origin: ascorbate (vitamin C), α-tocopherol (vitamin E) and xanthophylls. The only xanthophylls from dietary sources that accumulate in the retina are lutein and zeaxanthin [31,32,33,34,35,36,37]. They reach the greatest concentration in the retinal area occupied by axons of photoreceptors in the fovea, but a considerable amount, 15–25%, of retinal xanthophylls are present in the photoreceptor outer segments (POS) [38,39]. Sommerburg and colleagues extracted 27.6 and 9.02 ng of xanthophylls from human POS and RPE, respectively [38]. Considering the surface of the retina of 0.136 cm^2^ [40] and thickness of POS and RPE layer as 24 and 8 µm [41], respectively, the concentrations of xanthophylls can be estimated as 1.5 µM both in POS and RPE. Both lutein and zeaxanthin are excellent quenchers of excited states of photosensitizers and singlet oxygen. Zeaxanthin is more effective than lutein as a singlet oxygen quencher with the quenching rate constant of 12.6 × 10^9^ M^−1^ s^−1^, which is close to the diffusion-controlled limit [42,43,44]. The process of singlet oxygen quenching by xanthophylls is mainly physical, whereby carotenoids accept the excess of energy from singlet oxygen and deactivate to the ground state via internal conversion, with only a small fraction of interactions with singlet oxygen, less than 0.1%, leading to the oxidation of the carotenoid [45,46,47]. Xanthophylls, like other carotenoids, can also act as free radical scavengers; however, this mode of action can lead to the formation of reactive carotenoid radical and leads to their degradation [42,43,46].

Alpha-tocopherol quenches singlet oxygen but with rate constants 20 to 100 times smaller than that of zeaxanthin. The reported values range from 0.12 to 0.62 × 10^9^ M^−1^ s^−1^, with 2–7% of quenching resulting in the oxidation of α-tocopherol and formation of quinones and other products [48]. Alpha-tocopherol is a well-known free radical scavenger that can effectively inhibit lipid peroxidation, acting as a chain-breaking antioxidant [49]. It can also reduce semi-oxidized lutein, zeaxanthin and ATR to their original states [42,43,46,50]. Alpha-tocopherol is the main tocopherol present in human tissues, including the retina [38,51,52,53,54]. Friedrichson and colleagues determined that α- and γ-tocopherols content (80–84% of which was α-tocopherol) in human neural retinas and RPEs is about 24 nmol/retina, equivalent to 2.04 ± 0.62 nmol/mg protein, whereas the content in the RPE is 3.27 ± 1.48 nmol/mg protein [53]. This allows us to evaluate the concentrations of α-tocopherol in the neural retina to be about 67 µM (assuming the average thickness of the retina as 230 µm) and 101 µM in the RPE. Both values are two to three-fold greater than the average concentration of α-tocopherol in the blood plasma, which is about 25–35 µM [55,56]. 

Unlike carotenoids and tocopherols, ascorbate is a hydrophilic molecule that is soluble in water and negatively charged under physiological pH. Ascorbate is present in the human blood plasma and human vitreous at concentrations of about 50–80 µM and 2 mM, respectively [57,58]. The concentration of ascorbate in the blood plasma can be further increased up to 0.25 mM by oral supplementation [59]. Brain glial cells and neurons are particularly rich in ascorbate, where its concentrations are about 1 and 10 mM, respectively [60]. This accumulation is due to its active transport via sodium-dependent vitamin C transporters, which are highly expressed in the basal membrane of human RPE and with much smaller density in photoreceptors and other cells of the neural retina [57,58]. 

Ascorbate can quench singlet oxygen with a bimolecular rate constant of 0.3 × 10^8^ M^−1^ s^−1^ at pH 7.2, and this rate constant greatly decreases upon protonation [61]. As a result of interaction with singlet oxygen, hydrogen peroxide is formed, and ascorbate is oxidized to dehydroascorbate. Ascorbate can scavenge free radicals, including tocopheroxyl radical, and carotenoid and retinoid radical cations, thereby restoring these lipophilic antioxidants to the original state [42,43,46,49,50,62]. However, ascorbate can exert pro-oxidant effects in the presence of redox active metal ions such as iron by reducing these metal ions and thereby enabling them to function as catalysts of the Fenton reaction, where the reduced metal ion decomposes the hydrogen peroxide, leading to the formation of the most reactive free radical—the hydroxyl radical.

We have previously shown that the protection against photooxidative damage mediated by singlet oxygen and free radicals can be enhanced and the prooxidant effects of ascorbate in phototoxicity can be partly ameliorated, sometimes in a synergistic fashion, by using a combination of xanthophyll with α-tocopherol and/or ascorbate [63,64,65]. We have also shown that the mechanism responsible for the enhanced protection by zeaxanthin and another antioxidant that is a good free radical scavenger is based on the protection of zeaxanthin from oxidative degradation so that it can function longer as an unsurpassed singlet oxygen quencher. 

A similar combination of antioxidants was tested in a large clinical trial—Age-Related Eye Disease Study 2 (AREDS2)—on 4103 patients with age-related macular degeneration (AMD) [66]. The participants were supplemented daily with xanthophylls (10 mg lutein and 2 mg zeaxanthin), 400 IU vitamin E, 500 mg vitamin C, zinc oxide and cupric oxide with or without omega-3 lipids or β-carotene and followed up for a period of about 5 years over which 1940 eyes of 1608 patients progressed to advanced AMD. AREDS2 demonstrated that patients taking xanthophylls and vitamins C and E had a 9% (*p* = 0.05) reduced rate of AMD progression from the moderate to advanced form in comparison with patients not supplemented with xanthophylls [67]. The protective effect of xanthophyls was even more pronounced when compared between groups of patients supplemented with xanthophylls and no β-carotene and patients supplemented with β-carotene but no xanthophylls, where the risk of progression was reduced by 18% (*p* = 0.02) [68]. It needs to be noted that β-carotene, as a precursor of ATR, has the potential to increase its levels in the retina, thereby increasing the risk of (photo)toxicity [37].

Patients with the lowest dietary intake of macular xanthophylls benefited the most from xanthophyll supplementation, which reduced the risk of AMD progression by 26% (*p* = 0.01). The risk of AMD progression in patients with the highest dietary intake of macular xanthophylls was decreased by 10%, but the decrease was not statistically significant (*p* = 0.41) [67]. It is still unclear what the optimal levels of xanthophylls, vitamin E and vitamin C are in the retina. 

Therefore, the aim of this present study was to investigate in an in vitro model of the RPE-photoreceptor outer segments (POS) how different combinations of zeaxanthin, vitamin E and vitamin C can affect the cytotoxicity induced by photoexcited ATR. To mimic the situation in the retina in vivo, we used cultured ARPE-19 cells and liposomes containing unsaturated lipids and ATR. Cells and/or liposomes were enriched with lipophilic antioxidants, whereas ascorbate was added to PBS just before the exposure to light. The photooxidation of liposomes was monitored for selected combinations of antioxidants by oximetry. Our results show that supplementing cells with single lipophilic antioxidants has only a minor effect on the phototoxicity of ATR in liposomes. Combining α-tocopherol with zeaxanthin provides a statistically significant but still rather small protective effect. The incorporation of single antioxidants into liposomes also does not provide much protective effects to cells despite both zeaxanthin and tocopherol being able to effectively inhibit the initial rates of photooxidation. Combinations of zeaxanthin with α-tocopherol in liposomes and cells provide substantial protection enhancing cell viability from ~26% in the absence of antioxidants to ~63% in the presence of 4 µM zeaxanthin and 80 µM α-tocopherol, and this protective effect can be further increased to 69% in the presence of 0.5 mM ascorbate. Ascorbate on its own provides a dose-dependent protective effect up to 0.5 mM. At a concentration of 1 mM, the protective effect of ascorbate disappears, or is at most similar to that for 0.5 mM, whereas 2 mM of ascorbate exacerbates the phototoxicity. Zeaxanthin or α-tocopherol at high concentrations can enhance the protective effects of ascorbate or partly ameliorate its cytotoxic effects. 

## 2. Materials and Methods

### 2.1. General Chemicals and Reagents

Unless stated otherwise, all solvents, chemicals (including egg yolk phosphatidylcholine (EYPC), all-*trans*-retinaldehyde (ATR), 93-(4,5-dimethylthiazolyl-2)-2,5-diphenyltetrazolium bromide (MTT), butylated hydroxytoluene (BHT), Chelex 100) and cell culture consumables were obtained from Sigma-Aldrich Chemical Co. (St. Louis, MI, USA) or ThermoFisher (Waltham, MA, USA). Zeaxanthin was a generous gift from DSM Nutritional Products AG (Basel, Switzerland). The spin probe, 4-protio-3-carbamoyl-2,2,5,5-tetraperdeuteromethyl-3-pyrroline-1-yloxy (mHCTPO), was a generous gift from Professor Howard Halpern, University of Chicago, IL, USA. All procedures involving ATR and/or zeaxanthin, except for the experimental exposures to light, were performed under dim light.

### 2.2. Preparation of Liposomes

Multi-lamellar liposomes were prepared either from EYPC in the presence and absence of ATR, zeaxanthin and/or α-tocopherol [42,43,44]. The compounds dissolved in chloroform were mixed at required ratios in a round-bottom flask, and then the chloroform was evaporated under a stream of argon to form a lipid film, which was then kept under vacuum for at least 1 h. The lipid films were hydrated in nitrogen-saturated Dulbecco’s phosphate buffered saline with calcium and magnesium (PBS), aliquoted into cryovials and kept frozen until use. 

### 2.3. Comparison of the Effects of Ascorbate, α-Tocopherol and Zeaxanthin on Photosensitized Oxidation of Lipids

To determine the effect of antioxidants on photosensitized oxidation mediated by ATR, electron spin resonance (ESR) oximetry was used as described previously [42,44,45]. In short, oxygen concentrations were monitored by spectral characteristics of mHCTPO. The suspensions of liposomes containing 0.5 mM ATR, 7 mg/mL EYPC and selected single antioxidants or their combinations were irradiated in situ in the presence of 0.1 mM mHCTPO in a flat quartz cell (Wilmad Glass. Co., Vineland, NJ, USA), in a resonant cavity of the ESR spectrometer, at ambient temperature using a 150 W xenon arc lamp (Oriel Corporation, Stratford, CT, USA; 06,497 Model 60100) equipped with a combination of lenses and filters (a 5 g/L copper sulphate solution with 10 cm optical pathlength, a glass cut-off filter blocking light < 390 nm and an interference filter transmitting light of 404 ± 6 nm). The irradiance inside the resonant cavity, measured with a photodiode PD Irradiance Meter (Hamamatsu, Photonics, K.K., Hamamatsu City, Japan), was 6.2 mW/cm^2^. Other experimental conditions were as described previously [44].

### 2.4. Cell Culture

ARPE-19 cells, a spontaneously immortalized and well-characterized human retinal pigment epithelial cell line derived from a 19-year-old male donor [46], were purchased from the American Type Culture Collection (ATCC, Manassas, VA, USA). ARPE-19 cells were routinely cultured in 75 cm^2^ flasks, passaged by dissociation in 0.05% (*w*/*v*) trypsin and maintained at 37 °C in a humidified incubator filled with 5% CO_2_ in air. For experiments, ARPE-19 from passage numbers 27–29 were seeded in 24-well plates [19,47] and fed three times every 2–3 days with DMEM/F12 containing 10% heat-inactivated fetal calf serum (FCS), L-glutamine and penicillin–streptomycin (Sigma-Aldrich), which allowed the cells to become fully confluent. Then, for another three feedings, cells were fed with culture medium supplemented with 2% FCS.

### 2.5. Supplementation of Cells with Lipophilic Antioxidants

Stock solutions of 1 mM zeaxanthin, 20 mM α-tocopherol or their mixture were prepared under argon in DMSO under dim light. A day after the third feeding with DMEM/F12 containing 2% FCS and one hour before exposure to liposomes and light, 2 µL of stock solutions with the lipophilic antioxidants were injected directly into the wells, and plates were returned to the incubator. This provided the final concentrations of 2 μM zeaxanthin, 40 μM α-tocopherol and 0.2% DMSO. The high lipophilicity of both antioxidants was expected to cause their precipitation on the cell plasma membrane followed by their solubilization in the lipid bilayer [44]. Following the 1 h of incubation, wells with cells were washed with PBS to remove DMSO. We have shown previously that this approach allows for recovery from cells of 85% and 53% of supplemented zeaxanthin and/or α-tocopherol, respectively [64].

### 2.6. Exposure of Cells to Light and Liposomes

Just before the exposure, 0.5 mL of PBS (with or without ascorbate) and 0.5 mL of liposomal suspension was added to each well, and the plates were exposed to light or incubated in the dark for the pre-selected times. For the light exposure, the 24-well plates were placed on the top of a glass sheet above the set of fluorescent tubes, and cells were irradiated with white fluorescent light. The irradiance was measured with an IL2000 Spectrotube spectroradiometer (International Light Inc., Newbury, MA, USA), and the distance of the glass plate from the fluorescent tubes was adjusted to give irradiance at the plate level of 0.46 mW/cm^2^. The plates for incubation in the dark were covered by a cap made from blackened aluminum foil and placed in proximity to the light-exposed cells to ensure exposure to the same temperature as the irradiated cells. 

After the exposure to light or incubation in dark, liposomal suspensions were removed, cells were washed with PBS, and then PBS was replaced with the culture medium with 2% FCS, and the plate was returned to the incubator for 24 h, after which the assay of metabolic activity of cells was performed.

### 2.7. Evaluation of Cell Viability

The RPE cells were routinely observed using the inverted microscope. Cell viability was quantified by measurements of metabolic activity by the MTT assay as described before [19,44,47,48]. The effect of supplementation with lipophilic antioxidants on cell density was tested by counting cells in a Bürker chamber after trypsinization. For the MTT assay, cells were incubated for 60 min with 0.5 mg/mL MTT. Then, the cells were washed with PBS and solubilized with acidified isopropanol. The absorbance of the solubilized formazan was measured at 570 nm. The metabolic activities of cells are expressed relative to those of cells from the same plate supplemented with the same lipophilic antioxidants as cells exposed to ATR in EYPC liposomes with and without various combinations of lipophilic antioxidants [40].

### 2.8. Statistical Analysis

Unless stated otherwise, results are expressed as means ± SDs. Statistical analyses were performed using SigmaPlot14 and one or two-way ANOVA followed by all pairwise multiple comparison procedures using the Holm–Sidak method, where *p* < 0.05 were considered statistically significant.

## 3. Results

### 3.1. Effects of Vitamin C, Vitamin E and/or Zeaxanthin on Photosensitized Oxidation

We have shown previously that zeaxanthin can inhibit photosensitized oxidation when incorporated into liposomes together with ATR [44]. Here, we used a similar approach to compare the effects of zeaxanthin with α-tocopherol, ascorbate and their combinations (Figure 1). Liposomes containing 0.5 mM ATR and unsaturated lipids in the absence and presence of antioxidants were exposed to light to induce photosensitized oxidation, which was monitored by the measurement of oxygen consumption. The exposure of liposomes to light in the absence of antioxidants induced a rapid oxygen uptake, which increased by ~5% in the presence of 0.5 mM ascorbate, but the increase was not statistically significant (*p* = 0.269). In the presence of vitamin E, there was a dose-dependent inhibition of oxygen consumption, with its initial rate slowed down by ~19% and ~46% for 20 and 40 µM α-tocopherol, respectively (*p* < 0.001 in both cases). 

The addition of 0.5 mM ascorbate to liposomes with 40 µM α-tocopherol significantly increased oxygen uptake by ~17% in comparison with liposomes with 40 µM α-tocopherol in the absence of ascorbate (*p* = 0.049). Still, the mixture of 40 µM α-tocopherol with 0.5 mM ascorbate provided a significant inhibition of photooxidation, by ~37%, when compared with liposomes without antioxidants (*p* < 0.001). A mixture of 0.5 mM ascorbate with 4 µM zeaxanthin provided ~47% inhibition of photooxidation (*p* < 0.001), whereas 4 µM zeaxanthin alone was even more effective, providing ~49% inhibition (*p* < 0.001). The difference between the effects of 4 µM zeaxanthin alone and in combination with 0.5 mM ascorbate was not statistically significant (*p* = 0.593).

Decreasing zeaxanthin concentration two-fold to 2 µM diminished its protective effect and inhibited the rate of photooxidation by only ~21% (*p* < 0.001). A combination of 2 µM zeaxanthin with 20 µM α-tocopherol decreased the rate of photooxidation by ~41% in comparison with liposomes without antioxidants (*p* < 0.001).

Adding 0.5 mM ascorbate to that mixture increased the rate of photooxidation by ~6% in comparison with the absence of ascorbate, but this increase was not statistically significant (*p* = 0.396). The mixture of three antioxidants—0.5 mM ascorbate, 20 µM α-tocopherol and 2 µM zeaxanthin—provided a substantial ~38% inhibition of the photooxidation in comparison with samples without antioxidants (*p* < 0.001). 

These results indicate that vitamin C at a concentration of 0.5 mM does not offer a protective effect against photooxidation induced by photoexcited ATR. They also show that small—2 and 4 µM—concentrations of zeaxanthin can be similarly effective in inhibiting the photooxidation induced by photoexcited ATR as 10-fold greater concentrations of α-tocopherol (*p* = 0.616 and 0.41, respectively). Finally, the results demonstrate that a combination of 2 µM zeaxanthin with 20 µM α-tocopherol can offer a similar protection to doubling the concentration of a single lipophilic antioxidant (*p* = 0.078 and 0.322 for comparisons with 4 µM zeaxanthin and 40 µM α-tocopherol, respectively). 

### 3.2. Effects of Various Liposomal ATR Concentrations and Exposure Times in Dark and Light on RPE Cell Viability

The exposure of RPE cells to liposomes containing ATR led to the ATR concentration-dependent loss of cell viability, and the effect was substantially enhanced by increasing the incubation time and exposure to light (Figure 2). Sixty minutes of incubation with liposomes containing 0.5 mM ATR led to a relatively small ~16% decrease of cell viability for cells incubated in dark and a substantial ~74% decrease in cell viability for cells exposed to light. Therefore, 0.5 mM ATR and 60 min exposure to light were selected for further experiments testing the effects of antioxidants.

### 3.3. Effects on Phototoxicity of Supplementation of RPE Cells with Lipophilic Antioxidants

To determine if enriching cells with lipophilic antioxidants can ameliorate the phototoxicity induced by ATR in liposomes, cells were pre-treated 1 h before the exposure with zeaxanthin and/or α-tocopherol injected to the culture medium directly from its stock solution in DMSO. This approach was undertaken based on our previous results showing that the accumulation of zeaxanthin or other xanthophylls inside cells does not protect them from the phototoxicity of the photosensitizers that are external to the cells, including ATR in liposomes, whereas the supplementation of xanthophylls shortly before the exposure to photosensitizers and light, allowing for their precipitation on a cell plasma membrane followed by at least partial solubilization, does provide significant protection [44]. The removal of DMSO by washing cells in PBS before the addition of liposomes was needed because DMSO is a potent free radical scavenger and therefore could potentially introduce a non-physiological factor affecting the responses of cells to the treatments. 

The survival of cells supplemented with zeaxanthin increased to ~29%, which was a statistically significant increase in comparison with the ~26% survival of cells without added antioxidants (*p* < 0.001) (Figure 3). Supplementation with α-tocopherol increased cell survival to ~30%, and this increase was also statistically significant (*p* < 0.001) in comparison with not-supplemented cells. Supplementation with both α-tocopherol and zeaxanthin led to the greatest protection, with ~35% of cells surviving. This increase was statistically significantly different not only in comparison with cells without antioxidants but also in comparison with cells supplemented with either α-tocopherol or zeaxanthin (*p* < 0.001 in all three cases). 

### 3.4. Effects on Phototoxicity to RPE Cells of Zeaxanthin Supplementation and Incorporation of Zeaxanthin and α-Tocopherol into ATR Liposomes

The incorporation of increasing concentrations of zeaxanthin in liposomes with ATR did not substantially increase the non-supplemented cell survival after exposure to light in comparison with treatments without zeaxanthin (Figure 4A). When zeaxanthin was supplemented to cells and liposomes, only 4 µM of liposomal zeaxanthin significantly increased the cell viability from ~29% in the absence of liposomal zeaxanthin to ~34% (*p* < 0.001) (Figure 4A).

For cells not supplemented with antioxidants, increasing the concentration of zeaxanthin in liposomes with ATR and 10 or 20 µM α-tocopherol did not lead to any statistically significant increase in cell survival in comparison to treatments with liposomes without zeaxanthin but with the corresponding concentrations of α-tocopherol (Figure 4B). The addition of 0.5, 1 or 2 µM of zeaxanthin to liposomes with 40 µM of α-tocopherol and ATR also did not significantly increase the cell survival. However, the addition of 4 µM zeaxanthin to liposomes containing ATR and 40 µM of α-tocopherol increased cell viability from ~31% in the absence of zeaxanthin to ~40% in the presence of 4 µM of zeaxanthin (*p* < 0.001). When tested together with 80 µM of α-tocopherol, zeaxanthin exhibited a concentration-dependent protective effect with 2 µM of zeaxanthin increasing cell viability from ~35% to ~40% (*p* = 0.007), whereas with 4 µM of zeaxanthin, the cell viability increased to ~49% (*p* < 0.001).

The protection could be further increased by incorporating combinations of lipophilic antioxidants into liposomes, for cells supplemented with zeaxanthin (Figure 4C). For combinations of different concentrations of liposomal zeaxanthin with 10 µM of α-tocopherol, only 4 µM of zeaxanthin with 10 µM of α-tocopherol demonstrated a significant protective effect in comparison with liposomes without antioxidants, where the cell survival increased from ~29% to ~34% (*p* < 0.001). There was no statistical difference between the effect of 4 µM zeaxanthin alone and in combination with 10 µM α-tocopherol. 

Treatment of zeaxanthin-supplemented cells with combinations of 20 µM of α-tocopherol with different concentrations of zeaxanthin showed an increased protection only for 4 µM of zeaxanthin, which increased cell viability to ~37% (Figure 4C), and this was statistically significant not only in comparison with liposomes without antioxidants (*p* < 0.001) but also in comparison with liposomes with 20 µM of α-tocopherol without zeaxanthin (*p* = 0.006), for which the cell survival was ~32%. 

The treatment of zeaxanthin-supplemented cells with combinations of 40 µM of α-tocopherol and different concentrations of zeaxanthin showed increased cell survival rates to ~35%, 37% and 44% for 1, 2 and 4 µM zeaxanthin, respectively (Figure 4C), and these increases were statistically significant in comparison with liposomes without antioxidants (*p* < 0.001 in all three cases). The combination of 4 µM of zeaxanthin and 40 µM of α-tocopherol increased the cell survival also in comparison with liposomes containing 40 µM α-tocopherol but without zeaxanthin, for which the cell survival was ~32% (*p* < 0.001).

The treatment of zeaxanthin-supplemented cells with combinations of 80 µM α-tocopherol and different concentrations of zeaxanthin showed increased survival rates of 40%, 41%, 45% and 56% for 0.5, 1, 2 and 4 µM of zeaxanthin, respectively (Figure 4C), and these increases were statistically significant in comparison with liposomes without antioxidants (*p* < 0.001 in all four cases). These increases were also statistically significant in comparison with liposomes with 80 µM of α-tocopherol and without zeaxanthin (*p* = 0.031, 0.003, <0.001 and <0.001, respectively).

### 3.5. Effects on ATR Phototoxicity to RPE Cells of α-Tocopherol Supplementation and Incorporation of Zeaxanthin and α-Tocopherol into ATR Liposomes

The inclusion of α-tocopherol in liposomes provided a concentration-dependent increase in cell viability from ~26% to ~27%, 28%, 31% and 35% for concentrations of 10, 20, 40 and 80 µM, respectively, and achieved statistically significant differences in comparison with liposomes without tocopherol for the three highest concentrations (Figure 5A) (*p* = 0.019, *p* < 0.001 and *p* < 0.001, respectively). When the cells were supplemented with α-tocopherol, the addition of 10 or 20 µM of α-tocopherol into liposomes did not affect cell viability in comparison with liposomes without α-tocopherol, but 40 and 80 µM of liposomal α-tocopherol increased cell survival from ~30% in the absence of liposomal α-tocopherol to ~34% and 39% in the presence of 40 and 80 µM of α-tocopherol in liposomes (*p* < 0.001 in both cases). Comparisons of the survival of cells supplemented and not supplemented with α-tocopherol and exposed to liposomes with the same concentration of α-tocopherol showed that a statistically significant difference was achieved only for 80 µM α-tocopherol (*p* = 0.019). Liposomal α-tocopherol at a concentration of 80 µM provided similar protection in not-supplemented cells as 40 µM of α-tocopherol for cells supplemented with α-tocopherol (*p* = 0.783). The protection of 80 µM liposomal α-tocopherol for not-supplemented cells was significantly greater than that of 20 µM liposomal α-tocopherol for cells supplemented with α-tocopherol (*p* = 0.034).

Figure 5B shows the same data as Figure 4B plotted as a function of α-tocopherol concentration and is included to facilitate the comparison of the effects of various combinations of liposomal zeaxanthin and α-tocopherol on cells with and without supplementation with α-tocopherol. When cells were supplemented with α-tocopherol, combinations of antioxidants in liposomes provided some additional protection in comparison with cells without antioxidants, which was statistically significant for combinations of 1 µM zeaxanthin with 80 µM α-tocopherol (*p* = 0.023), 2 µM zeaxanthin with 40 µM α-tocopherol (*p* = 0.004), 2 µM zeaxanthin with 80 µM α-tocopherol (*p* = 0.011), 4 µM zeaxanthin with 40 µM α-tocopherol (*p* = 0.002) and 4 µM zeaxanthin with 80 µM α-tocopherol (*p* = 0.012) (Figure 5B,C). For cells supplemented with α-tocopherol, the greatest protection was observed for a combination of 4 µM zeaxanthin and 80 µM tocopherol, which together increased cell survival to ~54% (Figure 5C).

### 3.6. Effects of Different Combinations of Vitamin E and Zeaxanthin in Cells and/or in ATR-Containing Liposomes on Phototoxicity

The exposure of cells supplemented in both zeaxanthin and α-tocopherol to ATR-containing liposomes with zeaxanthin (Figure 6A) or α-tocopherol (Figure 6B) led to a further increase in cell viability, which was more pronounced for liposomes containing α-tocopherol than zeaxanthin. The survival of cells exposed to liposomes with 4 µM of zeaxanthin increased to ~39%, in comparison to liposomes without zeaxanthin, where the survival was ~35% (*p* = 0.033). Lower concentrations of liposomal zeaxanthin did not induce a statistically significant increase in the protective effect. The survival of cells exposed to liposomes with 80 µM of α-tocopherol increased to ~45% in comparison to liposomes without α-tocopherol, where the survival was ~35% (*p* < 0.001). Lower concentrations of liposomal α-tocopherol did not induce a statistically significant increase in the protective effect. 

Including various combinations of zeaxanthin and α-tocopherol in liposomes led to a further increase in cell viability, reaching ~63% for liposomes containing 4 µM of zeaxanthin and 80 µM of α-tocopherol (Figure 6C,D). The two-way analysis of variance showed that the effect of different levels of zeaxanthin depends on what level of α-tocopherol is present and that there was a statistically significant interaction between α-tocopherol and zeaxanthin (*p* ≤ 0.001).

### 3.7. Effects of Vitamin C on (Photo) Toxicity

The addition of Vitamin C during exposure to ATR and light led to a photoprotective effect, which became statistically significant only at 0.5 mM, whereas 2 mM ascorbate exerted a deleterious effect (Figure 7). The protection by ascorbate increased from ~26% in its absence to ~33% at 0.5 mM (*p* < 0.001), which was followed by a decrease to ~28% at 1 mM (*p* = 0.006 for comparison of the difference between the means for 0.5 and 1 mM of ascorbate; *p* = 0.137 for comparison for 0 and 1 mM of ascorbate). In the presence of 2 mM of ascorbate, the cell viability decreased to ~13% (*p* < 0.001 in comparisons with exposures in the absence of ascorbate and in the presence of all other ascorbate concentrations). The cytotoxic effect of ascorbate was not observed when cells were exposed to ascorbate and light in the absence of ATR or were incubated with ATR liposomes in the dark. 

### 3.8. Effects of Different Combinations of Vitamins C and E on Phototoxicity

When cells were exposed to light and ATR liposomes with α-tocopherol in the presence of different concentrations of ascorbate, the addition of ascorbate at concentration up to 0.5 mM further increased cell survival (Figure 8A). For combinations of ascorbate with 10 µM α-tocopherol, a statistically significant increase in the cell survival was only found for 0.5 mM of ascorbate (*p* < 0.001), where it increased from ~27% in the absence of ascorbate to ~35%. For combinations of ascorbate with 20 µM of α-tocopherol, there was no statistically significant increase in cell survival in comparison with cells in the absence of ascorbate. For combinations of ascorbate with 40 µM α-tocopherol, there was no statistically significant increase in cell survival for 0.1 mM ascorbate in comparison with cells in the absence of ascorbate, but for 0.2 and 0.5 mM of ascorbate, the cell survival increased to ~38% and 39%, respectively (*p* = 0.01 and 0.005, respectively). For combinations of ascorbate with 80 µM of α-tocopherol, there was no statistically significant increase in cell survival for 0.1 and 0.2 mM of ascorbate in comparison with cells in the absence of ascorbate, but for 0.5 mM of ascorbate, the cell survival increased to ~42% (*p* = 0.006). For 1 mM of ascorbate in combination with any α-tocopherol concentration, the cell survival was not different than that in the absence of ascorbate. Ascorbate at a 2 mM concentration increased phototoxicity for all α-tocopherol concentrations in comparison with treatments without ascorbate. Only in the presence of 80 µM of α-tocopherol was the phototoxic effect of ascorbate partly ameliorated, and cell survival increased to ~24% in comparison with ~13% in the absence of α-tocopherol (*p* < 0.001).

When cells were supplemented with α-tocopherol, ascorbate at a 0.5 mM concentration further increased its protective effect, but only in the presence of 40 or 80 µM of liposomal α-tocopherol, where the cell survivals were increased from ~24% and 39% to ~42% and 48%, respectively (*p* = 0.011 and *p* < 0.001) (Figure 8B). The cell survival in the presence of 1 mM ascorbate was similar as in its absence for all concentrations of α-tocopherol as well as in its absence (*p* < 0.001 for all five comparisons). When compared with 2 mM of ascorbate in the presence of liposomes without α-tocopherol, liposomes with 40 and 80 µM of α-tocopherol partly ameliorated the deleterious effect of ascorbate and increased cell survival to ~21% and 28% in comparison with ~14% in the absence of α-tocopherol (*p* = 0.047 and *p* < 0.001). 

Interestingly, the comparisons of the effects of the same combinations of ascorbate and liposomal α-tocopherol on cells with and without α-tocopherol showed no significant differences (Figure 8A,B).

Two-way analysis of variance demonstrated that the effect of different levels of α-tocopherol was dependent on what level of ascorbate was present, and there was a statistically significant interaction between α-tocopherol and ascorbate.

### 3.9. Effects of Different Combinations of Vitamins C and Zeaxanthin on Phototoxicity

When the cells were exposed to light and ATR in the presence of a combination of ascorbate and zeaxanthin, the protective effects were less pronounced than for combinations of α-tocopherol with ascorbate (Figure 8). While there was a trend of increased protection with increasing concentrations of ascorbate up to a 0.5 mM concentration, a statistically significant difference in comparison with treatment without ascorbate was achieved only in the case of 0.5 and 4 µM of liposomal zeaxanthin in the presence of 0.5 mM of ascorbate (*p* = 0.034 and *p* < 0.001, respectively). The combination of 4 µM of zeaxanthin with 0.5 mM of ascorbate was the most beneficial, increasing the cell viability to ~37% from ~28% in the presence of 4 µM of zeaxanthin and in the absence of ascorbate (*p* < 0.001) (Figure 8C). In the presence of zeaxanthin in liposomes, the effects of 1 mM of ascorbate were similar to those in the absence of ascorbate, whereas the deleterious effect of 2 mM of ascorbate was significant for all concentrations of liposomal zeaxanthin when compared with treatments without ascorbate and the same zeaxanthin concentration (*p* < 0.001 for all zeaxanthin concentrations). Liposomal zeaxanthin did not offer a statistically significant protection from the deleterious effects of ascorbate at any of the concentrations tested. 

When the cells were supplemented with zeaxanthin, the protective effects of ascorbate also peaked at 0.5 mM, increasing cell viability to ~37% from ~29% in the absence of ascorbate (*p* < 0.001), while the lower ascorbate concentrations did not lead to a statistically significant increase in cell survival (Figure 8D). In comparison with cells not supplemented with lipophilic antioxidants, where the cell survival was ~28%, 0.5 mM of ascorbate increased cell survival to ~37% in cells supplemented with zeaxanthin (Figure 8C,D). The phototoxicity to zeaxanthin-supplemented cells was similar in the presence of 1 mM of ascorbate to that in its absence. Ascorbate at a 2 mM concentration exacerbated the phototoxicity and significantly decreased cell survival in comparison with treatment without ascorbate (*p* < 0.001).

The protective effects of ascorbate for zeaxanthin-supplemented cells appeared to increase in the presence of liposomal zeaxanthin up to ~40% for a combination of 0.5 mM of ascorbate and 4 µM of liposomal zeaxanthin, but for no concentration of liposomal zeaxanthin was that effect statistically significant in comparison with treatment without liposomal zeaxanthin (Figure 8D). Comparisons of the effects of 0.5 mM of ascorbate for zeaxanthin-supplemented and not-supplemented cells did not show statistically significant differences for any concentration of liposomal zeaxanthin (Figure 8C,D).

The deleterious effect of 2 mM of ascorbate was significant for all concentrations of liposomal zeaxanthin when compared with treatments without ascorbate and the same zeaxanthin concentration (*p* < 0.001 for all zeaxanthin concentrations) (Figure 8D). The presence of zeaxanthin both in cells and in liposomes improved protection from the deleterious effects of 2 mM ascorbate, but only for 4 µM of liposomal zeaxanthin, which increased cell survival to ~22% in comparison with treatment in the absence of liposomal zeaxanthin (*p* < 0.001). 

A two-way analysis of variance demonstrated that the effect of different levels of zeaxanthin did not depend on the level of ascorbate present. There was not a statistically significant interaction between zeaxanthin and ascorbate.

### 3.10. Effects on Phototoxicity of Different Combinations of Three Antioxidants: Vitamins C, E and Zeaxanthin

Next, we evaluated a possibility that combinations of three antioxidants can offer increased protection compared to combinations of two antioxidants. In comparison with cells exposed to liposomes without α-tocopherol, 80 µM of α-tocopherol in liposomes induced a significant increase in protection for each combination of zeaxanthin with ascorbate (Figure 8C and Figure 9A). The maximal cell survival of ~54% was achieved with a combination of 4 µM zeaxanthin, 80 µM α-tocopherol and 0.5 mM ascorbate (Figure 9A). 

When the treatments in the presence and absence of ascorbate were compared, the addition of ascorbate at concentrations up to 0.5 mM showed a trend of increasing cell survival, but these increases were not statistically different to those in the absence of ascorbate for any concentration of zeaxanthin and 80 µM α-tocopherol (Figure 9A). Because of that, we did not test combinations of zeaxanthin and ascorbate with lower concentrations of α-tocopherol. 

In comparison with cells without ascorbate, 2 mM of ascorbate significantly exacerbated the phototoxicity to cells exposed to 80 µM of α-tocopherol and different concentrations of zeaxanthin (Figure 9A). In comparison with the survival of cells subjected to treatment without any antioxidants, a combination of 4 µM of zeaxanthin, 80 µM of α-tocopherol and 2 mM of ascorbate increased cell survival from ~26% to ~37% (*p* < 0.001), whereas treatments with lower concentrations of zeaxanthin in these combinations did not bring about any statistically significant increase in the survival of cells (Figure 3 and Figure 9A).

Finally, we tested combinations of three antioxidants on cells supplemented with both zeaxanthin and α-tocopherol (Figure 9B–D). When liposomes contained only zeaxanthin, increasing concentrations of ascorbate exerted a statistically significant protective effect only in cases of combinations of 0.5 mM of ascorbate with 1 and 4 µM of liposomal zeaxanthin, where the cell survivals increased from ~37% to ~42% and from ~40% to ~49%, respectively (*p* = 0.039 and *p* = 0.029, respectively) (Figure 9B). Combinations of 2 mM of ascorbate with any concentration of liposomal zeaxanthin led to increased phototoxicity when compared with the corresponding treatments without ascorbate (*p* < 0.001 for all concentrations of liposomal zeaxanthin). In addition, when compared with the phototoxicity to cells without any antioxidants, 2 mM of ascorbate exacerbated the toxicity for cells supplemented with zeaxanthin and α-tocopherol and exposed to liposomes containing any concentration of zeaxanthin except for 4 µM, for which the cell survival was similar to that of cells without any antioxidants (Figure 3; Figure 9B). None of liposomal zeaxanthin concentration provided a statistically significant protection from the toxicity of 2 mM of ascorbate.

When cells supplemented with zeaxanthin and α-tocopherol were exposed to ATR liposomes with 20 µM of α-tocopherol and different concentrations of zeaxanthin and ascorbate, there was a trend of increasing cell survival with increasing concentration of ascorbate up to the concentration of 0.5 mM, but it reached statistical significance only for 2 µM of zeaxanthin and 0.5 mM of ascorbate, where the survival increased to ~47% from ~41% for the same concentration of zeaxanthin in the absence of ascorbate (*p* = 0.016) (Figure 9C). The survival of cells in the presence of 4 µM of liposomal zeaxanthin and 0.5 mM of ascorbate was ~53%, but it was not statistically different than the ~46% survival rate in the absence of ascorbate at the same concentration of zeaxanthin (*p* = 0.895). There was no statistically significant difference in the absence and presence of 20 µM liposomal α-tocopherol for any corresponding treatments (Figure 9B,C).

When cells supplemented with zeaxanthin and α-tocopherol were exposed to ATR liposomes with 80 µM α-tocopherol and different concentrations of zeaxanthin and ascorbate, there was a trend of increasing cell survival with increasing concentration of ascorbate up to the concentration of 0.5 mM, but it did not reach statistical significance for any concentration of zeaxanthin (Figure 9D). Ascorbate at a 2 mM concentration exacerbated the toxicity in comparison with the corresponding treatments in the absence of ascorbate for all concentrations of liposomal zeaxanthin (*p* < 0.001) except for 1 µM (*p* = 0.568).

Comparisons of the effects of treatments of cells supplemented with zeaxanthin and α-tocopherol and exposed to ATR liposomes containing different concentrations of zeaxanthin in the absence and presence of 80 µM α-tocopherol showed a statistically significant protection for 80 µM of liposomal α-tocopherol in combinations with 0.5 µM of zeaxanthin and 0.5 mM of ascorbate (*p* = 0.01), 0.5 µM zeaxanthin and 2 mM ascorbate (*p* < 0.001), 1 µM of zeaxanthin and 0.1 mM of ascorbate (*p* = 0.037), 1 µM of zeaxanthin and 1 mM of ascorbate (*p* = 0.001), 1 µM of zeaxanthin and 2 mM of ascorbate (*p* < 0.001), as well as 2 and 4 µM of zeaxanthin with any concentration of ascorbate (*p* < 0.001 in all cases) (Figure 9B,D).

Comparisons of the effects of treatments of cells supplemented with zeaxanthin and α-tocopherol and exposed to ATR liposomes containing different concentrations of zeaxanthin in the presence of 20 and 80 µM of α-tocopherol showed a statistically significant protection for 80 µM of liposomal α-tocopherol in combinations with 1 µM of zeaxanthin and 2 mM of ascorbate (*p* < 0.001), as well as 2 and 4 µM of zeaxanthin with any concentration of ascorbate (*p* = 0.037 or lower) (Figure 9C,D). The most effective combination of three antioxidants was for cells supplemented with zeaxanthin and α-tocopherol and exposed to ATR liposomes containing 4 µM of zeaxanthin 80 µM of α-tocopherol in the presence of 0.5 mM of ascorbate, for which the cell survival increased to ~69%.

## 4. Discussion

### 4.1. Effects of Single Antioxidants

Our results demonstrate that single antioxidants, zeaxanthin, vitamin E or vitamin C provide rather limited protection from the phototoxic effects on ARPE-19 cells of ATR in liposomes as a model of POS.

The supplementation of cells with zeaxanthin increased cell survival from ~26% to only ~29%, despite the concentration of this antioxidant averaged over the cell monolayer likely exceeding its highest concentration in human RPE in vivo. In our experiments, the averaged concentrations of zeaxanthin in the ARPE-19 monolayer were ~1.1 mM—a concentration that is detected in the area of photoreceptor axons in the *macula lutea* part of the retina and three orders of magnitude greater than the average ~1.5 µM concentrations of xanthophylls detected in the human RPE and POS [38]. The incorporation of zeaxanthin in liposomes, in the absence of other antioxidants, did not provide any significant protection for cells without zeaxanthin and provided a significant protection only at the highest concentration tested of 4 µM for cells supplemented with zeaxanthin, where it increased cell survival to 34%. There was no significant protection offered by 2 µM liposomal zeaxanthin, which is close to its average concentration of ~1.5 µM in POS. It needs to be considered that the results of Sommerburg and colleagues were acquired before xanthophyll supplements were commercially available. The xanthophyll content varies greatly between different retinas and for most individuals can be increased via increased dietary intake and/or supplementation [69]. Bhosale and colleagues have determined that the total xanthophyll contents in macular retinas vary between individuals from ~1 ng up to ~115 ng and that the average retinal content of xanthophylls is almost three-fold increased in the supplemented in comparison with not-supplemented individuals [69]. They have also shown that the increase in xanthophyll content upon supplementation is similar in the macula and peripheral retina. Rapp and colleagues determined that xanthophyll content in the macula highly correlates with xanthophyl content in POS [39], suggesting that reaching 4 µM of zeaxanthin content may be achievable. It needs to be considered that xanthophyll accumulation can be dependent on the abundance of xanthophyll binding proteins and the activity of BCO2 proteins, which decompose xanthophylls [44,70,71]. 

The supplementation of ARPE-19 cells with vitamin E provided an averaged concentration in the cell monolayer of ~12 mM, which is about 100-fold higher than the concentration of 101 µM estimated based on results obtained from human RPE ex vivo reported by Friedrichson and colleagues [53]. Despite this high concentration, vitamin E increased cell survival from ~26% to only ~30%. Similar or greater protection could be achieved by the incorporation of α-tocopherol in liposomes where 20, 40 and 80 µM of α-tocopherol increased cell survival to ~28%, 31% and 35%, respectively. This shows that α-tocopherol can achieve the protective effect even at concentrations below its average retinal concentrations of 67 µM [53]. It needs to be considered that the range of tocopherol concentrations detected in the human neural retinas varies and can be as low as ~0.3 nmol/mg protein [53], corresponding to 10 µM, and that concentration is not protective against phototoxicity. In most cases, however, the tissue levels of α-tocopherol can be modulated by dietary intake and supplementation [72,73]. 

In addition to singlet oxygen quenching and the scavenging of free radicals derived from lipid peroxidation, α-tocopherol can scavenge ATR cation radicals, which can be formed by ATR interaction with peroxyl radicals, thereby preventing its degradation and the formation of more phototoxic products [50,74,75]. 

When the cells were supplemented with α-tocopherol, 40 µM of liposomal α-tocopherol provided similar protection to 80 µM of liposomal α-tocopherol in not-supplemented cells, whereas 80 µM of α-tocopherol increased the survival of supplemented cells to ~39%. 

Vitamin C provided a significant protection only at a 0.5 mM concentration, where it increased cell survival from ~26 to ~33%. The protective effect disappeared at a 1 mM concentration, whereas 2 mM of vitamin C decreased cell survival to ~13%. The vitreous concentration of ascorbate is 2 mM [76]. There are no diffusion barriers between the vitreous and retina, and therefore the 2 mM concentration can be expected to be the physiological level of ascorbate at the POS–RPE interface. We have previously observed deleterious effects of ascorbate where it exacerbated the light-induced oxidation of human melanosomes and RPE cell homogenates and exacerbated the phototoxic effect of rose bengal even at lower concentrations of 1.4, 0.7 and 0.35 mM [65,76,77,78]. However, a substantial number of experiments on animals point to the protective role of ascorbate against light-induced injury to the retina [62,79,80,81,82,83,84,85,86,87,88,89]. The deleterious effect of ascorbate may be due to the quenching of singlet oxygen by ascorbate, which results in the oxidation of ascorbate to dehydroascorbate and production of hydrogen peroxide [61]. Unlike singlet oxygen, whose lifetime in aqueous solutions is very short, and which therefore is unlikely to diffuse from liposomes to cells, hydrogen peroxide can diffuse to cells and can be decomposed by metal ions, such as iron, in their reduced state, leading to the oxidation of the metal ion and production of the most reactive free radical—the hydroxyl radical. Ascorbate can reduce the oxidized iron ion back to the state where it can decompose another molecule of hydrogen peroxide. The depletion of the total ascorbate as well as the increased ratio of the oxidized to the reduced form of ascorbate in the neural retina and the RPE–choroid complex, was observed in guinea pigs immediately after 6 h exposure to 10–20 klx of white fluorescent light, which produced mild damage to the RPE and photoreceptors [89]. It remains to be established how the retina protects itself from the deleterious effects of ascorbate, especially the AMD retina, which contains increased levels of iron in the RPE, including iron that is easily chelatable [21], while benefitting from its protective effects.

### 4.2. Effects of Combinations of Two Antioxidants: Zeaxanthin and α-Tocopherol

Combinations of zeaxanthin with α-tocopherol exerted beneficial effects both when supplemented to cells and/or incorporated into liposomes. When cells were enriched in zeaxanthin and α-tocopherol, the cell survival increased to ~35% from ~26% in the absence of antioxidants, whereas supplementation with single zeaxanthin or α-tocopherol increased cell survival to only ~29% and ~30%, respectively. The incorporation of 4 µM of zeaxanthin and 80 µM of α-tocopherol into liposomes increased the cell survival to ~49% and ~63% in not-supplemented cells and cells supplemented in both antioxidants, respectively. It is clear from the present study that the quenching of singlet oxygen and scavenging of free radicals needs to happen at the source of their production, which in this case were liposomes. While the concentrations of the lipophilic antioxidants providing protective effects are above the average concentrations of zeaxanthin and α-tocopherol in the human retina, they are still within the range present in the tissues examined and can be reached in most people by increasing dietary intake and/or supplementation. 

The protective effects of combinations of zeaxanthin with α-tocopherol are consistent with our previous studies showing that incorporation of α-tocopherol and zeaxanthin into liposomes or their supplementation to cells can offer a synergistic protection in comparison to the effects induced by zeaxanthin or α-tocopherol used as single antioxidants [63,64,65]. We have shown in previous studies on liposomes and cells that in photosensitized oxidation induced by photoexcited rose bengal and merocyanine 540, respectively, zeaxanthin was the primary antioxidant inhibiting oxidation by singlet oxygen, whereas α-tocopherol was more efficient than zeaxanthin in inhibiting oxidation mediated by free radicals [63,64]. The combination of both antioxidants slowed down not only the oxidation of lipids but also the consumption of each of them, thereby allowing each of them to provide their antioxidant protection for longer. Following these studies, other authors also reported beneficial effects of a combination of α-tocopherol and zeaxanthin, including preserving phagocytosis in lipofuscin-, melanosome- or melanolipofuscin-laden ARPE-19 cells exposed to sublethal levels of blue light [90,91,92].

### 4.3. Effects of Combinations of Two Antioxidants: Ascorbate with Zeaxanthin or α-Tocopherol

Combinations of α-tocopherol with ascorbate demonstrated an increase in protection mainly for 0.5 mM of ascorbate in comparison with treatments with only one of these antioxidants. Combinations of 0.5 mM of ascorbate with 80 µM of liposomal α-tocopherol increased cell survivals to ~42% and ~48% for not-supplemented cells and cells supplemented with α-tocopherol, respectively. For comparison, in the absence of ascorbate, the cell survivals were ~35% and ~39% for treatments with 80 µM of liposomal α-tocopherol for not-supplemented cells and cells supplemented with α-tocopherol, respectively. Cell survival after treatment with 0.5 mM ascorbate was ~33%.

Combinations of zeaxanthin with ascorbate brought about a substantial increase in protection only for a combination of 0.5 mM of ascorbate and 4 µM of liposomal zeaxanthin, increasing cell viability to ~37% and 40% from ~28% and 34% in the presence of 4 µM zeaxanthin and in the absence of ascorbate for cells without and with zeaxanthin supplementation, respectively.

Clearly, α -tocopherol can not only enhance the photoprotective effects of ascorbate better than zeaxanthin but is also more efficient in ameliorating the deleterious effects of 2 mM of ascorbate. Alpha-tocopherol at an 80 µM concentration in liposomes increased cell viability from ~13% in its absence to ~24% and 28% for not-supplemented cells and cells supplemented with α-tocopherol, respectively. Zeaxanthin at an 80 µM concentration in liposomes, on the other hand, did not significantly affect cell survival in non-supplemented cells and increased cell survival to ~22% for zeaxanthin-supplemented cells. These results together with the lack of interaction of zeaxanthin and ascorbate shown by the two-way ANOVA suggest that zeaxanthin does not efficiently protect ascorbate from oxidation by singlet oxygen and ascorbate does not efficiently protect zeaxanthin from free-radical mediated degradation, despite it having been reported that ascorbate can scavenge cation radicals of lutein and zeaxanthin in liposomal membranes [93].

It appears that ascorbate can provide additional protection in combination with α-tocopherol by the recycling of α-tocopherol from its tocopheroxyl radical [62]. While a similar recycling mechanism exists for zeaxanthin, which can be recycled by ascorbate from its semi-oxidized state [42,43,46], this interaction seems to play a minor role, if any, under our experimental conditions. This can be due to the limited access of ascorbate to zeaxanthin cation radical in the liposomal membrane and/or the zeaxanthin radical cations not being the main pathway of zeaxanthin degradation. 

### 4.4. Effects of Combinations of Three Antioxidants: Ascorbate with Zeaxanthin and α-Tocopherol

For not-supplemented cells, the addition of 0.5 mM of ascorbate during exposure to ATR liposomes with 4 µM of zeaxanthin and 80 µM of α-tocopherol provided a cell survival of ~54%, whereas for cells supplemented with zeaxanthin and α-tocopherol, the cell survival was even greater, reaching ~69%. While the addition of 0.5 mM ascorbate to the treatments with combinations of liposomal zeaxanthin and α-tocopherol showed a trend of increasing cell survival, it did not reach statistical significance. The relatively small effect of ascorbate on increasing cell survival in the presence of α-tocopherol and zeaxanthin is surprising, considering its potential to recycle both α-tocopherol and zeaxanthin, and deserves further investigation.

Importantly, the combination of 4 µM of zeaxanthin and 80 µM of α-tocopherol not only ameliorated the toxicity of 2 mM of ascorbate but increased the cell survival above that in the absence of any antioxidants. In comparison with the survival of cells subjected to treatment without any antioxidants, a combination of 4 µM of zeaxanthin, 80 µM of α-tocopherol and 2 mM of ascorbate increased cell survival from ~26% to ~37% for not-supplemented cells and to 41% for cells supplemented with zeaxanthin and α-tocopherol (*p* < 0.001 in both cases), thereby providing an explanation for how photoreceptors and RPE can avoid the deleterious effects of ascorbate. 

### 4.5. Physiological Relevance

The physiological relevance of this study needs to be considered. While we ensured that we tested antioxidants at physiologically relevant concentrations, there are several limitations to our model of the RPE–POS interface. While the release of ATR from dark-adapted POS, with concentrations of rhodopsin exceeding 3 mM, can easily exceed by several fold the 0.5 mM concentration used in this study, POS contain enzymes, namely ABCR and RDH8, that can rapidly reduce it to all-*trans*-retinol. On the other hand, it needs to be considered that these enzymes rely on ATP and NADPH as co-factors. The characteristic features of the aged retina, and the AMD retina in particular, are convoluted and mismanaged photoreceptor outer segments (POS) [94]. Therefore, it is likely that this can cause the impairment of ATR trafficking in the POS discs on its way to ABCR, which is present mostly in the rims of these discs, as well as the delivery of ATP and NADPH to the site of enzyme action. 

An important finding from our investigation is that deactivating the reactive oxygen species at the source of their formation, in this case liposomes, is more important than enriching cells in antioxidants. This can be due to the diffusion of cytotoxic end-products of lipid oxidation such as reactive carbonyls against which zeaxanthin and vitamins E and C are ineffective. A particularly important difference between POS and our liposomes is the lipid composition. We used EYPC, which is rich in monounsaturated oleic acid (C18:1) and linoleic acid (C18:2) and contains some arachidonic acid (C20:4), accounting for 27, 21 and 5 wt %, respectively [95]. POS contain a much greater contribution from unsaturated fatty acids, where the predominant unsaturated fatty acid is docosahexaenoic acid (C22:6) accounting for 30–35% of total fatty acids, followed by C18:1 (10–14%), and C20:4 (7–11%) [96]. Therefore, antioxidants in POS compete for reactive oxygen species with a much greater abundance of unsaturated double bonds. It needs to be considered that zeaxanthin, α-tocopherol and ascorbate are not the only antioxidants present in POS, which contain other low-molecular weight antioxidants as well as antioxidant and detoxification enzymes, including glutathione peroxidases, which can decompose hydrogen peroxide and lipid hydroperoxides [97,98,99].

Our results point to a potential deleterious effect of ascorbate at a 2 mM concentration, which is present in the vitreous [76] and therefore likely to be the same in the area of the POS and RPE interface. Our results also demonstrate that α-tocopherol and a combination of α-tocopherol and zeaxanthin can prevent these deleterious effects and even provide a protective effect by increasing cell survival from 26% in the absence of tested antioxidants to 41% for cells supplemented with α-tocopherol and zeaxanthin and exposed to 2 mM of ascorbate and ATR liposomes with 4 µM of zeaxanthin and 80 µM of α-tocopherol. The levels of ascorbate in tissues and plasma are tightly controlled by intestinal absorption and ascorbate transporters, which are responsible for its accumulation against its concentration gradient in many tissues [57,59]. AREDS2 supplement, which is meant to be taken daily, includes 500 mg of vitamin C, which is expected to provide a steady-state ~70 µM concentration of ascorbate in blood plasma, which is close to that achievable from five servings of fruit and vegetables [59].

Our results demonstrate a remarkably beneficial effect of the supplementation of cells and liposomes with zeaxanthin and α-tocopherol, where the most effective concentrations in liposomes were 4 µM for zeaxanthin and 80 µM for α-tocopherol, both of which are achievable by dietary intake. A question is whether increasing these concentrations by supplementation can further increase their protective effects. It has been shown that both zeaxanthin and α-tocopherol can exert pro-oxidant and cytotoxic effects [100,101,102,103]. Alpha-tocopherol can exert a pro-oxidant effect upon the interaction of a lipid-derived peroxyl radical and the formation of an α-tocopheroxyl radical, which can propagate lipid peroxidation by abstracting hydrogen from an unsaturated lipid, albeit with a smaller rate than is achieved by lipid-derived peroxyl radicals [101,103]. This is the reason why the regeneration of α-tocopherol from the α-tocopheroxyl radical, which can be achieved by ascorbate, is important in preserving its antioxidant potential and preventing its pro-oxidant action [49].

Chucair and colleagues have shown that lutein and zeaxanthin exert a protective effect against the oxidant-induced apoptosis of cultured rat photoreceptors in a narrow range of concentrations, where the maximal protection is provided by 170 and 140 nM for zeaxanthin and lutein, respectively, whereas higher concentrations of these xanthophylls cause deleterious effects [100]. While these low concentrations were stated, the volume in which the xanthophylls were provided or the actual concentrations present in the cells were not given; therefore, their results cannot be compared with the results presented here. The cytotoxic effects of lutein and zeaxanthin could be due to the prooxidant action of these xanthophylls, which has been reported for carotenoids at high oxygen tensions or at high concentrations of carotenoids, where they can form addition radicals with lipid-derived peroxyl radicals and propagate lipid peroxidation (reviewed in [46]). Alternatively, semi-oxidized carotenoids can be damaging to biomolecules such as tyrosine or cysteine [104]. The products of the oxidative degradation of these xanthophylls have been shown to cause loss of ARPE-19 cell viability and induce the generation of reactive oxygen species and apoptosis [102]. Two of these products, namely 3-hydroxy-β-ionone and 3-hydroxy-14′-apocarotenal, both likely derived from oxidative cleavage of lutein or zeaxanthin, have been identified in human retinas *post mortem* [105]. Therefore, increasing by supplementation the concentrations of lipophilic antioxidants, such as vitamin E, lutein or zeaxanthin, needs to be performed with caution. 

## 5. Conclusions

In conclusion, we have demonstrated that combinations of antioxidants of dietary origin—vitamin C, vitamin E and zeaxanthin—at physiologically relevant concentrations can offer a synergistic protection of ARPE-19 cells from the phototoxicity induced by ATR in liposomes. Our results also show that quenching singlet oxygen and/or scavenging free radicals close to the source of their generation—that is, in liposomes—is more effective than including antioxidants in cells. The different combinations of zeaxanthin with vitamin E and combinations of vitamin E with vitamin C exhibit interactions between these two pairs of antioxidants. No interaction could be detected for different combinations of vitamin C with zeaxanthin. The greatest protection was achieved for cells supplemented with α-tocopherol and zeaxanthin and exposed to ATR liposomes with 80 µM of α-tocopherol and 4 µM of zeaxanthin in the presence of 0.5 mM of ascorbate, where the cell survival was 69% in comparison with 26% in the absence of antioxidants. The concentrations of 80 µM of α-tocopherol and 4 µM of zeaxanthin were the highest concentrations of these antioxidants tested in this study and at the top concentrations detected in the human photoreceptor outer segments. Increasing the concentration of ascorbate to 2 mM caused a decrease in protection in the presence of high concentration of lipophilic antioxidants and enhanced the ATR phototoxicity when their concentrations were low. 

## Figures and Tables

**Figure 1 antioxidants-11-01132-f001:**
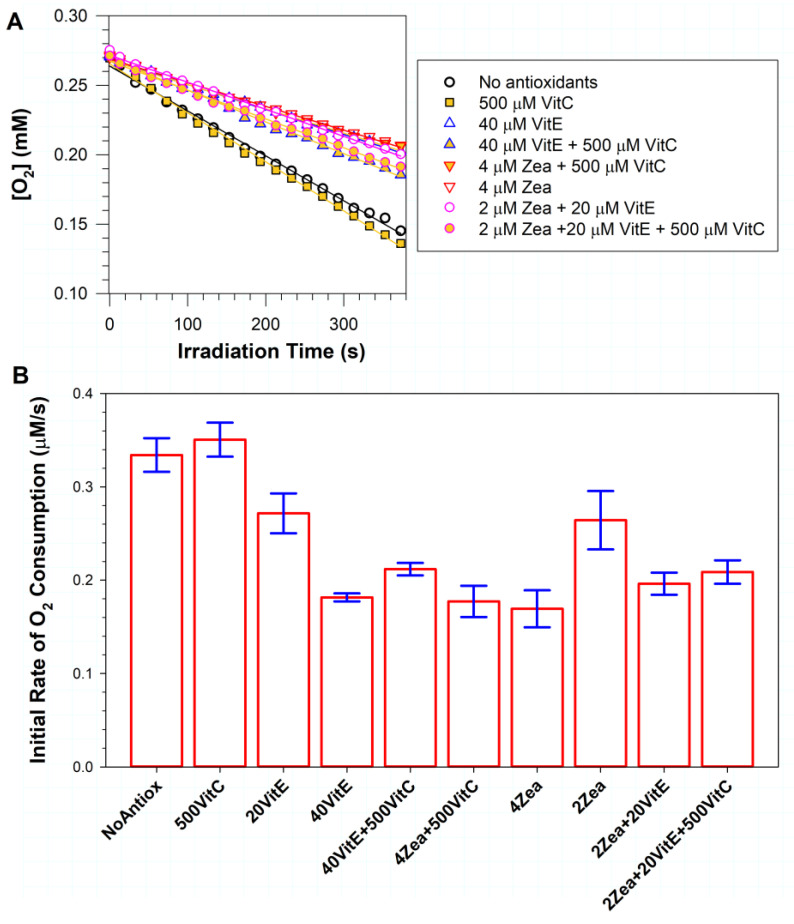
The effects of vitamin C (VitC), vitamin E (α-tocopherol, VitE), zeaxanthin (Zea) and their combinations on photooxidation induced by ATR. (**A**) Representative kinetics of oxygen consumption during exposure of ATR-containing liposomes to blue light in the absence and presence of antioxidants at indicated concentrations in µM. Suspension of liposomes, consisting of 7 mg/mL EYPC, 0.5 mM ATR and indicated concentrations of lipophilic antioxidants, in the absence or presence of 0.5 mM vitamin C were exposed to blue light. The samples included 0.1 mM mHCTPO used as a spin probe. The initial linear portions of the kinetics were fitted to straight lines, and the slopes of these lines gave the initial rates of oxygen consumption. (**B**) The initial rates of oxygen consumption in the absence(NoAntiox) and presence of indicated antioxidants. The heights of the bars indicate the means, while the error bars indicate SDs.

**Figure 2 antioxidants-11-01132-f002:**
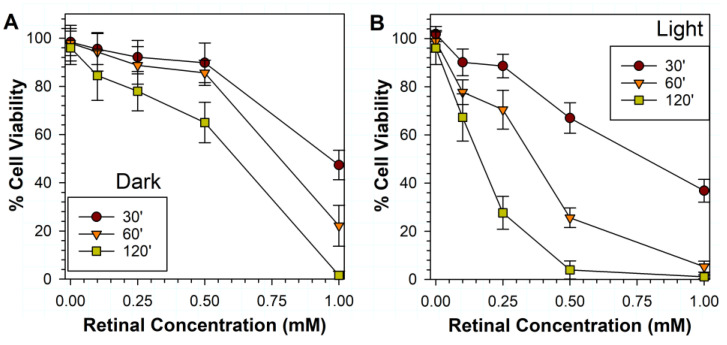
Phototoxicity to ARPE-19 cells of liposomes with different concentrations of ATR in the dark (**A**) and after exposure to visible light (**B**). The incubation times (in minutes) in the dark and during exposure to visible light are indicated in the figure legends. Cell viability was measured by MTT assay 24 h after the exposure. The symbols indicate the means, while the error bars indicate SDs.

**Figure 3 antioxidants-11-01132-f003:**
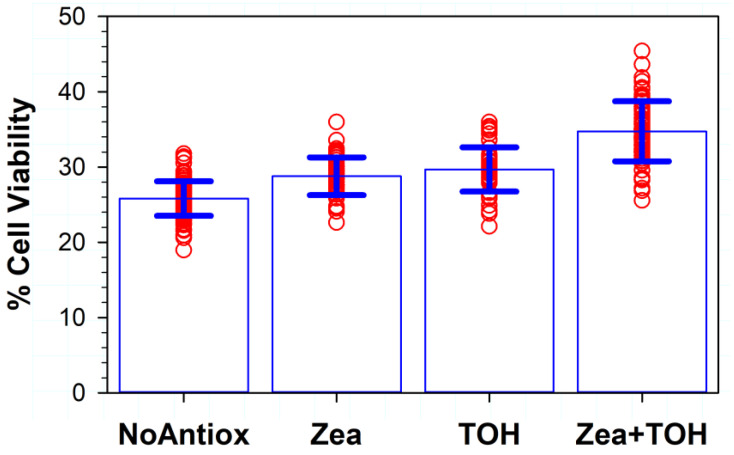
The effect of the pre-treatment of cells with lipophilic antioxidants vitamin E and/or zeaxanthin on the viability of ARPE-19 cells after 1 h exposure to visible light in the presence of liposomes containing 0.5 mM ATR. Cell viability was measured by MTT assay 24 h after the exposure. The symbols represent viability of cells in independent experiments, the heights of the bars indicate the means, while the error bars indicate SDs (N = 96 for experiments with no antioxidants (NoAntiox); N = 48 for experiments on cells supplemented with zeaxanthin (Zea) or α-tocopherol (TOH), N = 66 for experiments on cells supplemented with both zeaxanthin and α-tocopherol (Zea + TOH)).

**Figure 4 antioxidants-11-01132-f004:**
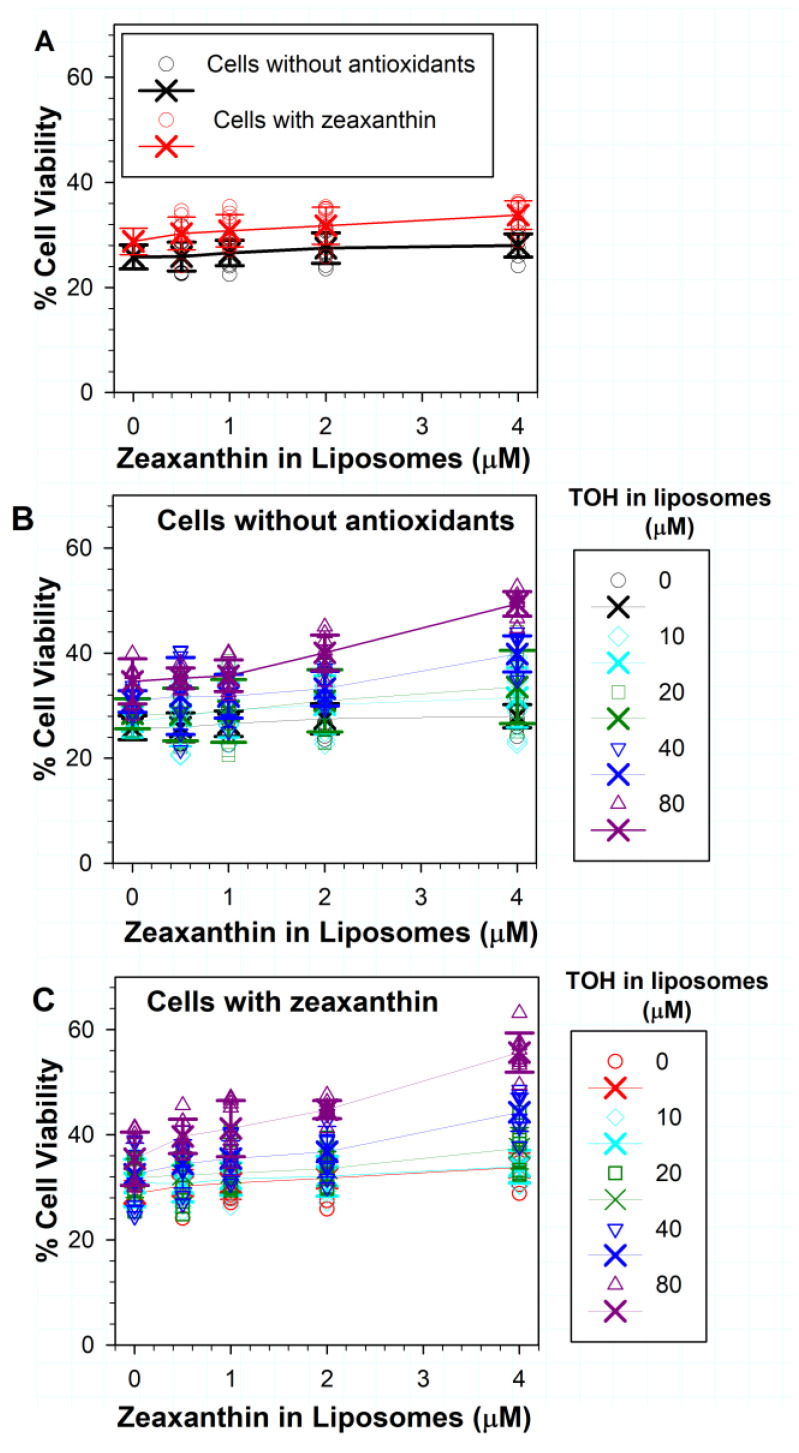
(**A**) Effects on ATR phototoxicity of different concentrations of zeaxanthin in liposomes to ARPE-19 cells supplemented with zeaxanthin and without zeaxanthin. (**B**) Effects on phototoxicity of different combinations of zeaxanthin and α-tocopherol (TOH) in liposomes to ARPE-19 cells not supplemented with antioxidants. (**C**) Effects on phototoxicity of different combinations of zeaxanthin and α-tocopherol in liposomes to ARPE-19 cells supplemented with zeaxanthin. The symbols represent the viability of cells in independent experiments, the crosses represent the means, while the error bars show SDs (the individual data points from experiments with no antioxidants in liposomes are shown only in Figure 3; N for each experiment with antioxidants in liposomes was 8). Other experimental conditions as in Figure 3.

**Figure 5 antioxidants-11-01132-f005:**
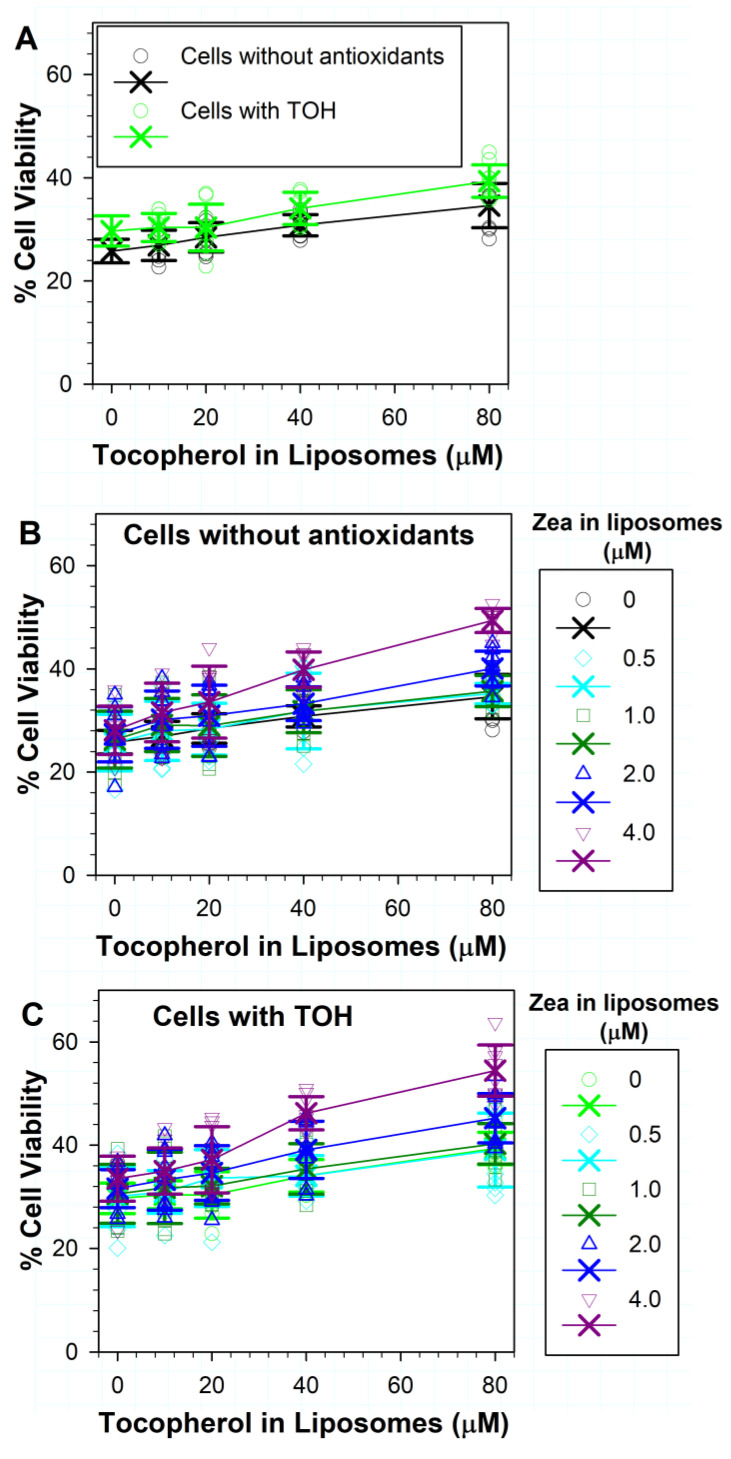
(**A**) Effects on phototoxicity of different concentrations of α-tocopherol in liposomes to ARPE-19 cells supplemented with α-tocopherol (TOH) and without α-tocopherol. (**B**) Effects on phototoxicity of different combinations of zeaxanthin (Zea) and α-tocopherol in liposomes to ARPE-19 cells not supplemented with antioxidants (the data are the same as in Figure 4B, plotted here as a function of α-tocopherol concentration). (**C**) Effects on phototoxicity of different combinations of zeaxanthin and α-tocopherol in liposomes to ARPE-19 cells supplemented with α-tocopherol. The symbols represent the viability of cells in independent experiments, the crosses represent the means, while the error bars show SDs (the individual data points from experiments with no antioxidants in liposomes are shown only in Figure 3; N for each experiment with antioxidants in liposomes was 8). Other experimental conditions as in Figure 3.

**Figure 6 antioxidants-11-01132-f006:**
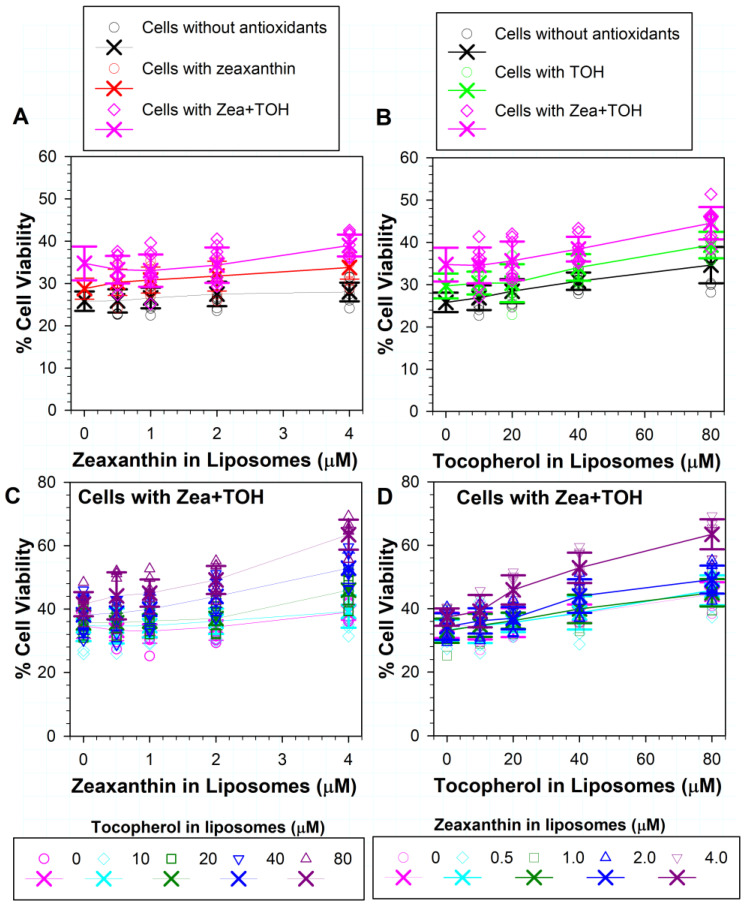
(**A**) Effects of different concentrations of zeaxanthin in liposomes on phototoxicity to ARPE-19 cells not supplemented with antioxidants, supplemented with zeaxanthin, or supplemented with a combination of α-tocopherol with zeaxanthin (Zea + TOH). (**B**) Effects of different concentrations of α-tocopherol (TOH) in liposomes on phototoxicity to ARPE-19 cells not supplemented with antioxidants, supplemented with α-tocopherol or supplemented with a combination of α-tocopherol with zeaxanthin (Zea + TOH). (**C**) Effects on phototoxicity of different combinations of zeaxanthin and α-tocopherol in liposomes to ARPE-19 cells supplemented with α-tocopherol and zeaxanthin. (**D**) The data are the same as in Figure 6C plotted as a function of α-tocopherol concentration. The symbols represent the viability of cells in independent experiments, the crosses represent the means, while the error bars show SDs (the individual data points from experiments with no antioxidants in liposomes are shown only in Figure 3; N for each experiment with antioxidants in liposomes was 8). Other experimental conditions as in Figure 3.

**Figure 7 antioxidants-11-01132-f007:**
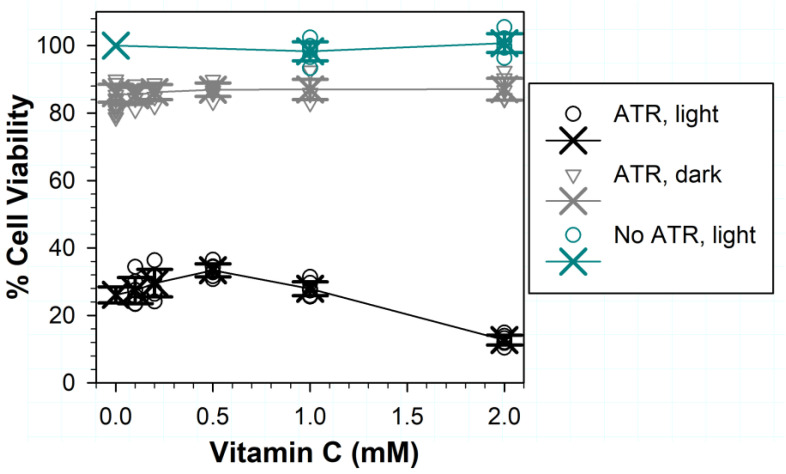
Effects on ARPE-19 cell viability of different concentrations of vitamin C for 1 h exposure to visible light (ATR, light) or incubation in the dark (ATR, dark) in the presence of liposomes containing 0.5 mM of ATR and 7 mg/mL of EYPC. Control cells in light-exposed plates were exposed to vitamin C and EYPC liposomes in the absence of ATR (No ATR, light). Cell viability was measured by MTT assay 24 h after the exposure. The symbols represent data from individual experiments, crosses represent the means, while the error bars represent SDs.

**Figure 8 antioxidants-11-01132-f008:**
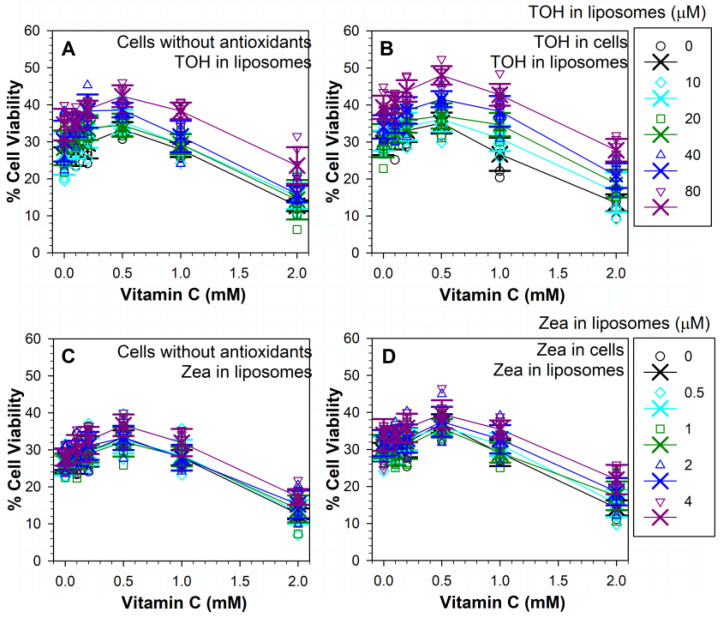
Effects of different concentrations of vitamin C in combination with α-tocopherol (TOH) (**A**,**B**) or zeaxanthin (Zea) (**C**,**D**) on the viability of ARPE-19 cells after 1 h exposure to visible light in the presence of EYPC liposomes containing 0.5 mM ATR and indicated concentrations of α-tocopherol or zeaxanthin. Cells were either not pre-treated with antioxidants (**A**,**C**) or supplemented with α-tocopherol (**B**) or zeaxanthin (**D**). Cell viability was measured by MTT assay 24 h after the exposure. The symbols represent data from individual experiments, crosses represent the means, while the error bars show SDs. The individual data points from experiments with no antioxidants in liposomes are shown only in Figure 3; N for experiments with antioxidants in liposomes and without ascorbate was 8; N for for experiments in the presence of ascorbate was 6). Other experimental conditions as in Figure 3.

**Figure 9 antioxidants-11-01132-f009:**
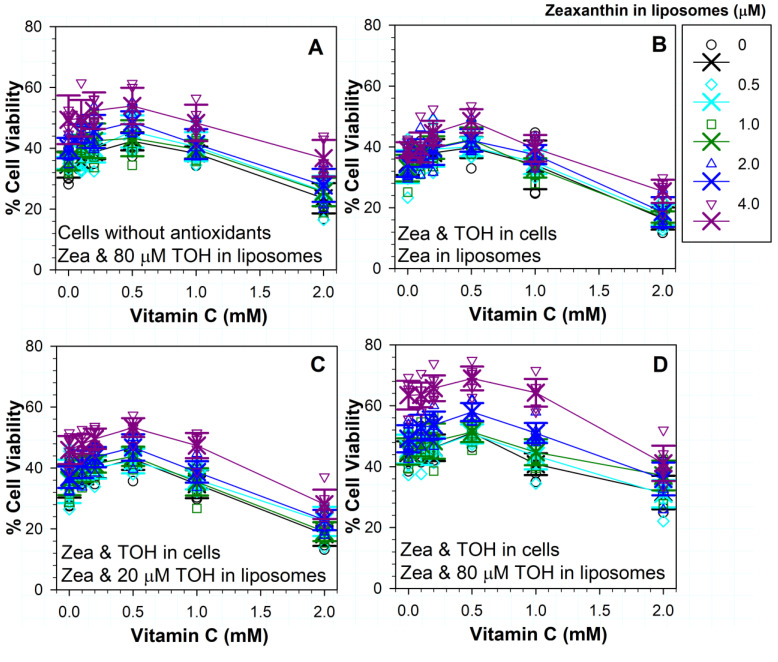
Effects of different combinations of three antioxidants—vitamin C, vitamin E (TOH) and zeaxanthin (Zea)—on the survival of ARPE-19 cells after exposure to visible light and liposomes containing 0.5 mM ATR with and without antioxidants. (**A**) Cells not supplemented with antioxidants, liposomes contained 80 µM α-tocopherol and different concentrations of zeaxanthin indicated in the legend. (**B**–**D**): cells supplemented with α-tocopherol and zeaxanthin liposomes contained the indicated concentrations of zeaxanthin in the absence (**B**) and presence of α-tocopherol at concentrations of 20 µM (**C**) and 80 µM (**D**). Cell viability was measured by MTT assay 24 h after the exposure. The symbols represent data from individual experiments, crosses represent the means, while the error bars show SDs. The individual data points from experiments with no antioxidants in liposomes are shown only in Figure 3; N for experiments with antioxidants in liposomes and without ascorbate was 8; N for experiments in the presence of ascorbate was 6). Other experimental conditions as in Figure 3.

## Data Availability

All data are presented in the manuscript.

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
