# Peer review of "Is There an Optimal Combination of AREDS2 Antioxidants Zeaxanthin, Vitamin E and Vitamin C on Light-Induced Toxicity of Vitamin A Aldehyde to the Retina?"

_antioxidants, 2022, doi:10.3390/antiox11061132_

Round 1

Reviewer 1 Report

The MS by Rozanowska and others deals with the protective effects of a combination of antioxidants on an in vitro retinal model.

The MS pretends to demonstrate the optimal antioxidant concentration of a combination of antioxidants that are included in the AREDS2 study. From the very beginning the abstract does not present the results in a clear way. The title does not clearly correlate with the experimental model presented and the results obtained.

The figures are not clearly designed to support how the results are written. The physiological relevance is a central point in this study, as the authors have well introduced, and needs to be clearly improved.

The authors have an extensive experience in the field and tend to cite a bit too much of their own work, missing relevant work of others. 

The MS can be clearly improved and be presented in a  much shorter format, with a different title and reduced number of self citations, and including missing authors relevant in the field.

Author Response

Response to Reviewer 1

We thank the Reviewer for reading the manuscript and their constructive criticism. Each point is addressed below.

The MS by Rozanowska and others deals with the protective effects of a combination of antioxidants on an in vitro retinal model.

The MS pretends to demonstrate the optimal antioxidant concentration of a combination of antioxidants that are included in the AREDS2 study. From the very beginning the abstract does not present the results in a clear way. The title does not clearly correlate with the experimental model presented and the results obtained.

The abstract clearly states the model we used to answer the question given in the title and summarises the most important findings. We discuss the physiological relevance of these findings in the manuscript, which makes the title relevant, and now we have added that information as a statement at the end of the abstract.

The figures are not clearly designed to support how the results are written. The physiological relevance is a central point in this study, as the authors have well introduced, and needs to be clearly improved.

Each section of the results initially compares the data within a single graph. Later in each section, the data from different graphs are compared and it is clearly indicated in brackets at which figures to look at. We kept the figure format similar to facilitate these comparisons. We have expanded the Discussion section on physiological relevance by adding three paragraphs (highlighted in yellow).

The authors have an extensive experience in the field and tend to cite a bit too much of their own work, missing relevant work of others. 

This study is building on our previous work and we feel citing it was necessary. All our work that was cited is highly relevant to the manuscript. When discussing the physiological relevance of the results we have added 11 references of other authors:

  1. Shui, Y.B.; Holekamp, N.M.; Kramer, B.C.; Crowley, J.R.; Wilkins, M.A.; Chu, F.; Malone, P.E.; Mangers, S.J.; Hou, J.H.; Siegfried, C.J.; et al. The gel state of the vitreous and ascorbate-dependent oxygen consumption: relationship to the etiology of nuclear cataracts. Arch Ophthalmol 2009, 127, 475-482, doi:10.1001/archophthalmol.2008.621.
  2. Olchawa, M.M.; Furso, J.A.; Szewczyk, G.M.; Sarna, T.J. Lipofuscin-mediated photic stress inhibits phagocytic activity of ARPE-19 cells; effect of donors' age and antioxidants. Free Radic. Res. 2017, 51, 799-811, doi:10.1080/10715762.2017.1380307.
  3. Olchawa, M.M.; Szewczyk, G.M.; Zadlo, A.C.; Krzysztynska-Kuleta, O.I.; Sarna, T.J. The effect of aging and antioxidants on photoreactivity and phototoxicity of human melanosomes: An in vitro study. Pigment Cell Melanoma Res. 2020, doi:10.1111/pcmr.12914.
  4. Olchawa, M.M.; Szewczyk, G.M.; Zadlo, A.C.; Sarna, M.W.; Wnuk, D.; Sarna, T.J. The Effect of Antioxidants on Photoreactivity and Phototoxic Potential of RPE Melanolipofuscin Granules from Human Donors of Different Age. Antioxidants 2020, 9, doi:10.3390/antiox9111044.
  5. Burke, M.; Edge, R.; Land, E.J.; Truscott, T.G. Characterisation of carotenoid radical cations in liposomal environments: interaction with vitamin C. J Photochem Photobiol B 2001, 60, 1-6, doi:10.1016/s1011-1344(01)00111-7.
  6. Chucair, A.J.; Rotstein, N.P.; Sangiovanni, J.P.; During, A.; Chew, E.Y.; Politi, L.E. Lutein and zeaxanthin protect photoreceptors from apoptosis induced by oxidative stress: relation with docosahexaenoic acid. Invest Ophthalmol Vis Sci 2007, 48, 5168-5177, doi:10.1167/iovs.07-0037.
  7. Firsov, A.M.; Franco, M.S.F.; Chistyakov, D.V.; Goriainov, S.V.; Sergeeva, M.G.; Kotova, E.A.; Fomich, M.A.; Bekish, A.V.; Sharko, O.L.; Shmanai, V.V.; et al. Deuterated polyunsaturated fatty acids inhibit photoirradiation-induced lipid peroxidation in lipid bilayers. J Photochem Photobiol B 2022, 229, 112425, doi:10.1016/j.jphotobiol.2022.112425.
  8. Kalariya, N.M.; Ramana, K.V.; Srivastava, S.K.; van Kuijk, F.J. Carotenoid derived aldehydes-induced oxidative stress causes apoptotic cell death in human retinal pigment epithelial cells. Exp Eye Res 2008, 86, 70-80, doi:10.1016/j.exer.2007.09.010.
  9. Lamberson, C.R.; Xu, L.; Muchalski, H.; Montenegro-Burke, J.R.; Shmanai, V.V.; Bekish, A.V.; McLean, J.A.; Clarke, C.F.; Shchepinov, M.S.; Porter, N.A. Unusual kinetic isotope effects of deuterium reinforced polyunsaturated fatty acids in tocopherol-mediated free radical chain oxidations. Journal of the American Chemical Society 2014, 136, 838-841, doi:10.1021/ja410569g.
  10. Burke, M.; Edge, R.; Land, E.J.; McGarvey, D.J.; Truscott, T.G. One-electron reduction potentials of dietary carotenoid radical cations in aqueous micellar environments. FEBS Lett 2001, 500, 132-136, doi:10.1016/s0014-5793(01)02601-1.
  11. Prasain, J.K.; Moore, R.; Hurst, J.S.; Barnes, S.; van Kuijk, F.J. Electrospray tandem mass spectrometric analysis of zeaxanthin and its oxidation products. J Mass Spectrom 2005, 40, 916-923, doi:10.1002/jms.868.

The MS can be clearly improved and be presented in a  much shorter format, with a different title and reduced number of self citations, and including missing authors relevant in the field.

The last three points have been addressed above. As for the shortening of the manuscript, we made the results description more concise where possible but apart from that change we have not found any part, which is not highly relevant and could be removed.

Reviewer 2 Report

General comments:

In the AREDS2 clinical trial, vitamin C, vitamin E and zeaxanthin were important supplements in preventing the progression of age-related macular degeneration. The purpose of this manuscript is to determine the effects of an optimal combination of vitamin C, vitamin E, and zeaxanthin on vitamin A aldehyde (ATR)-induced phototoxicity in retinal pigment epithelial cells (ARPE-19). The authors demonstrated that the combination of zeaxanthin (4 uM) and vitamin E (80 uM) in liposomes and ARPE-19 cells provided significant protection and enhanced cell viability compared to the absence of antioxidants.  Moreover, the addition of vitamin C (0.5 mM) provided further protection in ARPE-19 cells.  Although some of the data are interesting, there are major and minor issues needed to be addressed.

Comments to the authors:

Major comments:

  1. In this manuscript, they used ARPE-19 cells, an immortalized human retinal pigment epithelial (RPE) cell line. Immortalized cell line may not represent the phenotype of RPE cells under physiological conditions. Could the authors address this question using primary RPE cells from rats or mice?
  2. In this manuscript, they show that vitamin C provides protection at 0.5 mM, while 2 mM vitamin C causes deleterious effects in ARPE-19 cells. These results suggest that increasing the dose of vitamin C supplements may have side effects in humans or those with AMD. The authors need to discuss how this result can be applied to clinical applications.
  3. The main finding was that the combination of zeaxanthin (4 uM), vitamin E (80 uM) and vitamin C (0.5 mM) was the best formulation to protect ARPE-19 cells. The authors need to discuss how to apply this formula to clinical applications.

Minor comments:

  1. On line 42, the authors need to change “Absorption of light by visual pigments causes photoisomerization----“ to “Light absorption by visual pigments causes photoisomerization----.“
  2. On line 52, the authors need to change “Both singlet oxygen and superoxide are reactive oxygen species which can exert damage----“ to “Both singlet oxygen and superoxide are reactive oxygen species that can exert damage----.“
  3. On line 54, the authors need to change “biomolecules of the photoreceptor outer segment and neighbouring RPE cells” to “biomolecules of the photoreceptor outer segment and neighboring RPE cells.”
  4. On line 69, the authors need to change “Photoreceptors and RPE contain a number antioxidants” to “Photoreceptors and RPE contain a number of antioxidants.”
  5. On Line 70, the authors need to change “The only xanthophylls of dietary origin which---“ to “The only xanthophylls of dietary source which---“

Author Response

Response to Reviewer 2

We thank the Reviewer for reading the manuscript and their constructive criticism. Each point is addressed below.

General comments:

In the AREDS2 clinical trial, vitamin C, vitamin E and zeaxanthin were important supplements in preventing the progression of age-related macular degeneration. The purpose of this manuscript is to determine the effects of an optimal combination of vitamin C, vitamin E, and zeaxanthin on vitamin A aldehyde (ATR)-induced phototoxicity in retinal pigment epithelial cells (ARPE-19). The authors demonstrated that the combination of zeaxanthin (4 uM) and vitamin E (80 uM) in liposomes and ARPE-19 cells provided significant protection and enhanced cell viability compared to the absence of antioxidants.  Moreover, the addition of vitamin C (0.5 mM) provided further protection in ARPE-19 cells.  Although some of the data are interesting, there are major and minor issues needed to be addressed.

Comments to the authors:

Major comments:

  1. In this manuscript, they used ARPE-19 cells, an immortalized human retinal pigment epithelial (RPE) cell line. Immortalized cell line may not represent the phenotype of RPE cells under physiological conditions. Could the authors address this question using primary RPE cells from rats or mice?

Rats and mice are not a good model to study effects of lutein and zeaxanthin because they do not accumulate these xanthophylls in their retinas. ARPE-19 cells are well characterized and widely used as a model of human retinal pigment epithelial cells.

2. In this manuscript, they show that vitamin C provides protection at 0.5 mM, while 2 mM vitamin C causes deleterious effects in ARPE-19 cells. These results suggest that increasing the dose of vitamin C supplements may have side effects in humans or those with AMD. The authors need to discuss how this result can be applied to clinical applications.

As suggested, we have addressed that in the revised manuscript by adding a paragraph in the Discussion under Physiological relevance (highlighted in yellow).

3. The main finding was that the combination of zeaxanthin (4 uM), vitamin E (80 uM) and vitamin C (0.5 mM) was the best formulation to protect ARPE-19 cells. The authors need to discuss how to apply this formula to clinical applications.

As suggested, we have addressed that in the revised manuscript by adding a paragraph in the Discussion under Physiological relevance (highlighted in yellow).

Minor comments:

  1. On line 42, the authors need to change “Absorption of light by visual pigments causes photoisomerization----“ to “Light absorption by visual pigments causes photoisomerization----.“

Amended as suggested

2. On line 52, the authors need to change “Both singlet oxygen and superoxide are reactive oxygen species which can exert damage----“ to “Both singlet oxygen and superoxide are reactive oxygen species that can exert damage----.“

Amended as suggested

3. On line 54, the authors need to change “biomolecules of the photoreceptor outer segment and neighbouring RPE cells” to “biomolecules of the photoreceptor outer segment and neighboring RPE cells.”

We have used British English throughout the manuscript so we have kept British spelling in this instance as well.

4. On line 69, the authors need to change “Photoreceptors and RPE contain a number antioxidants” to “Photoreceptors and RPE contain a number of antioxidants.”

Amended as suggested

5. On Line 70, the authors need to change “The only xanthophylls of dietary origin which---“ to “The only xanthophylls of dietary source which---“

Amended as suggested

Round 2

Reviewer 1 Report

Although the shortening of the MS has not been extensive, the MS can be accepted.

Reviewer 2 Report

No further comments on it.